# A Mother’s Dilemma: The 5-P Model for Vaccine Decision-Making in Pregnancy

**DOI:** 10.3390/vaccines11071248

**Published:** 2023-07-17

**Authors:** Elizabeth Cox, Magali Sanchez, Katherine Taylor, Carly Baxter, Isabelle Crary, Emma Every, Brianne Futa, Kristina M. Adams Waldorf

**Affiliations:** 1Department of Health Systems and Population Health, School of Public Health, University of Washington, Seattle, WA 98195, USA; 2Department of Epidemiology, School of Public Health, University of Washington, Seattle, WA 98195, USA; 3Department of Obstetrics and Gynecology, University of Washington, Seattle, WA 98195, USA; 4School of Medicine, University of Washington, Seattle, WA 98195, USAeevery@uw.edu (E.E.); 5School of Medicine, Tulane University, New Orleans, LA 70112, USA; 6Department of Global Health, University of Washington, Seattle, WA 98195, USA

**Keywords:** pregnancy, vaccine, vaccine hesitancy, COVID-19, social media, rural medicine

## Abstract

Pregnant women are a highly vaccine-resistant population and face unique circumstances that complicate vaccine decision-making. Pregnant women are also at increased risk of adverse maternal and neonatal outcomes to many vaccine-preventable diseases. Several models have been proposed to describe factors informing vaccine hesitancy and acceptance. However, none of these existing models are applicable to the complex decision-making involved with vaccine acceptance during pregnancy. We propose a model for vaccine decision-making in pregnancy that incorporates the following key factors: (1) perceived information sufficiency regarding vaccination risks during pregnancy, (2) harm avoidance to protect the fetus, (3) relationship with a healthcare provider, (4) perceived benefits of vaccination, and (5) perceived disease susceptibility and severity during pregnancy. In addition to these factors, the availability of research on vaccine safety during pregnancy, social determinants of health, structural barriers to vaccine access, prior vaccine acceptance, and trust in the healthcare system play roles in decision-making. As a final step, the pregnant individual must balance the risks and benefits of vaccination for themselves and their fetus, which adds greater complexity to the decision. Our model represents a first step in synthesizing factors informing vaccine decision-making by pregnant women, who represent a highly vaccine-resistant population and who are also at high risk for adverse outcomes for many infectious diseases.

## 1. Introduction

Pregnant women and their fetuses are at greater risk for adverse outcomes of many infectious diseases compared to non-pregnant people, leading to an increased risk for hospitalization, maternal death, preterm birth, congenital anomalies, and stillbirth [1,2,3,4,5,6,7,8,9,10,11,12]. Vaccination can play a crucial role in protecting the health of pregnant women and fetuses due to maternal antibody transfer across the placenta and through breast milk after birth. The U.S. Centers for Disease Control recommend that women receive several vaccines during pregnancy, including vaccines for pertussis, tetanus, diphtheria, polio, influenza, and coronavirus disease of 2019 (COVID-19) [13]. Despite these recommendations, pregnant individuals are highly vaccine-hesitant. During the COVID-19 pandemic, vaccination rates for COVID-19 remained lower among pregnant versus non-pregnant women, even though contracting COVID-19 during pregnancy conferred a greater risk for adverse outcomes [14,15]. A significant knowledge gap exists regarding the factors influencing vaccine decision-making in pregnancy.

Barriers to vaccine uptake for pregnant individuals include safety concerns regarding the vaccine, lack of information on vaccines from care providers, perceived lack of data on the safety of vaccines, and low self-perceived risk of infection or severity of illness [16]. Despite the existence of many clinical practice guidelines regarding SARS-CoV-2 vaccination in pregnancy, there was an information gap between experts in pregnancy vaccination and the pregnant lay public [17]. Historically, pregnant women have also been excluded from clinical vaccine trials, which contributes to a lack of confidence as to the safety of vaccines for them or their fetuses [18,19]. Perceived insufficiency of data regarding vaccine safety for the fetus is a common reason for vaccine hesitancy among pregnant women [16,20]. An information gap and lack of research and public health messaging around vaccines and the concerns of pregnant or reproductive-aged women has led to disinformation linking vaccination to adverse fertility or pregnancy outcomes. To address these barriers, we launched a limited public health campaign and research program, called One Vax Two Lives, during the COVID-19 pandemic [21,22]. However, without a vaccine-decision-making model in pregnancy, a public health campaign is likely to miss key factors driving uptake.

Several health decision and health behavior models have been applied to vaccine hesitancy and acceptance, including the Health Belief Model [23,24,25], 3C [26], 4C [27], and the 5A models [28]. Pregnancy represents an intense information-seeking phase in life, in which pregnant women face heightened awareness of the impact of their health decisions on their fetuses; the sensitivity of this decision also increases vulnerability to disinformation. The unique complexity of pregnancy is not captured by any of the general models of health behavior or vaccine hesitancy. Our objective is to highlight the challenges of vaccine decision-making in pregnancy and to present a new vaccine-acceptance model that can be used to inform public health campaigns and provider–patient interactions targeting pregnant women. We begin by exploring barriers to vaccination in pregnancy. Secondly, we address current vaccine-acceptance models and the ways in which they can and cannot be applied to pregnant individuals as decision-makers. Finally, we propose a new 5-P model for vaccine decision-making in pregnancy and address how these factors influence vaccination and other health decisions of pregnant individuals.

## 2. Barriers to Vaccination in Pregnancy

The recent COVID-19 pandemic and rapid rollout of vaccines presented an opportunity to understand barriers to vaccination in pregnancy. Vaccine hesitancy can be generally divided into structural and attitudinal barriers, with the latter playing a larger role in why pregnant individuals have greater vaccine hesitancy [29]. Early reasons for COVID-19 vaccine hesitancy involved the concern about the lack of data regarding its safety during pregnancy [30,31,32,33]. For pregnant individuals, vaccine safety was initially unknown, as pregnant and lactating individuals were not included in the initial Phase 3 studies [18]. One study found that 98.7% of pregnant respondents were aware that there was a COVID-19 vaccine available but very few were knowledgeable about the vaccine [31]. Furthermore, the available information about vaccination in pregnancy was often contradictory, making it difficult to know what was accurate and further enhancing skepticism about the vaccine’s efficacy and safety [34]. The internet and social media were significant sources of misinformation, perpetuating confusion and vaccine hesitancy. Concern about the COVID-19 vaccine’s efficacy was also a reason for vaccine hesitancy in pregnant and breastfeeding individuals [31,33,34]. In addition to pregnancy-specific and other attitudinal barriers to vaccination in pregnancy, there are also many other structural barriers to vaccination (e.g., access, affordability, trust in science), which are addressed in many vaccine-acceptance models designed for the general population.

## 3. Models of Vaccine Acceptance

There are no specific models for vaccine acceptance in pregnancy, but several constructs that explain factors informing vaccine hesitancy and acceptance exist; many of these models build upon the Health Belief Model (HBM) [23,24,25], which was generally designed to describe factors informing health decision-making. The HBM is divided into several categories: perceived susceptibility to the disease, perceived severity of the disease, perceived benefits of the disease-prevention behavior, barriers to the disease-prevention behavior, a cue to action, and self-efficacy. The cue to action represents events, people, or things that help trigger a change in health behavior. Self-efficacy refers to one’s confidence in the health behavior, which could represent one’s trust and confidence that vaccination is a healthy choice.

In 2014, a new vaccine-decision-making model was proposed, called the 3C model. The 3C model describes three factors of vaccine hesitancy: confidence, complacency, and convenience [26]. Confidence can be described as a belief in the safety of vaccines, the reliability of the healthcare systems that deliver them, and the motivations of policymakers. Complacency can be described as belief (or lack thereof) in the necessity of the vaccine with respect to other life responsibilities. Convenience can be described as the access to and affordability of the vaccine for those who want it. This was expanded in 2015 in the 4C model to include a utility calculation, where the benefits of the vaccine play a larger role [27]. In 2016, the 5A model was developed, in which vaccine acceptance was described in terms of access, affordability, awareness, acceptance, and activation [28]. The factors, strengths, and limitations are described with respect to vaccination and vaccination in pregnancy in Table 1.

These models operate on similar decision-making factors and principles: access, affordability, confidence or self-efficacy, complacency, and utility of intervention. Historic and continual research on the social determinants of health has demonstrated a vast number of factors that influence health outcomes outside of a healthcare setting. Therefore, decision-making frameworks should be heavily based on factors that are contextualized within an individual’s social environment. Access, affordability, and complacency are all factors whose primary effects occur before the patient even decides to make an appointment, and they are arguably the most important factors in determining whether a patient makes a positive healthcare decision. A provider has little power to instill confidence in and utility of healthcare preventions and interventions if a patient lacks access to the provider or does not think to go to the provider. While these factors are helpful and accurate for the general population of the United States, they are less applicable to pregnant individuals, who have more frequent interactions with their healthcare providers during the duration of their pregnancy than non-pregnant adults.

Although there are no specific models for vaccine acceptance in pregnancy, a model has been proposed to describe decision-making in pregnancy that may be applied to vaccine acceptance [35]. This model describes three themes in the decisions of pregnant people: uncertainty, bodily autonomy, and being a good mother. These three themes are interlinked with three actions: information gathering, balancing aspects of a choice, and aligning with a birth philosophy. This model is also inspired by the knowledge that choices during pregnancy cannot be simplified to weighing the pros and cons of medical actions. Instead, it is impacted by perceptions of birth as a natural process or a medical process, desires to maintain body autonomy and independent choice, and stressors of pregnancy [35]. Application of this model to vaccine decision-making addresses, in part, many of the limitations of more traditional decision-making models by describing specifically how safety and perceived worth impact health decisions in pregnancy. However, vaccine decision-making in pregnancy is even more complex, which we describe in a new model.

## 4. The 5-P Vaccine Acceptance Model in Pregnancy

Although the models in Table 1 are easily applied to the general population, pregnant women have a more complex decision-making process that generally involves balancing risks and benefits for themselves and the fetus [36,37]. We now propose a 5-P model of vaccine acceptance in pregnancy that combines our clinical experience, features of the HBM, and a pregnancy decision-making model [23,24,35,36,37]. By drawing on these models [25,35], literature review as a research methodology [38], as well as our experience from the One Vax Two Lives COVID-19 research and public health campaign [21,22], we can address key elements unique to pregnancy decision-making.

The basic tenets of the HBM certainly remain true in that health decisions are influenced by a combination of one’s perceptions of the benefits and barriers related to that health behavior as well as perceptions of disease susceptibility and severity. The pregnant population intersects with this model, but carries additional distinctive factors separate from the rest of the general population; therefore, we propose a new model of vaccine decision-making that is unique to the pregnant perspective.

However, additional elements should be considered that are unique to pregnancy, including the intense information-seeking nature of pregnancy, the frequent exposure to healthcare providers that can impart their knowledge of and opinions on vaccination during pregnancy, and a frequent stance of harm avoidance that is typical in pregnancy. Historically, there has been little scientific evidence regarding the safety and efficacy of new vaccines in pregnancy, as pregnant people were excluded. Furthermore, public health campaigns have not prioritized the dissemination of health information to pregnant people, as they represent a minority of the population. Consequently, at a time in life when information about health decisions is valued highly, there has been little to no information available on the safety and benefits of a new vaccine during pregnancy.

Many other modifying factors can also influence an individual’s perceptions and, ultimately, their health behavior. These factors include characteristics of an individual, such as their age, geographic location, income, or level of education. For example, rural areas in the United States are associated with lower vaccination uptake among pregnant people than non-rural areas [39]. Race and ethnicity are other modifying factors. Historically, Black and Latinx people have shown greater vaccine hesitancy and lower vaccine uptake. This can be accredited to barriers to health behavior created by racism and a history of harm perpetuated by the medical field on these communities. A 2020 Kaiser Family Foundation survey found that 45% of Black patients had experienced negative experiences with a healthcare provider [40]. Vaccine decision-making in pregnancy clearly exists on the background of a person’s socioeconomic, geographic, educational, and racial/ethnic identity.

The key factors informing the 5-P model include: (1) perceived information sufficiency on vaccine safety in pregnancy, (2) protection of pregnancy (harm avoidance), (3) provider–patient relationship, (4) perceived vaccine benefits for mother and fetus, and (5) perceived disease susceptibility and severity in pregnancy (Figure 1, Table 2). Two themes regarding decision-making in the pregnancy model, uncertainty and bodily autonomy, ultimately play into the “perceived information sufficiency” and “balancing pros and cons for mother and fetus” elements of our model, respectively. Additional elements that are more universal in medical decision-making also contribute to vaccine decision-making in pregnancy and can be positive, negative, or neutral factors in the ultimate decision. The final step in the process requires a careful balancing of the risks and benefits for oneself and one’s fetus, which can be a taxing process given the many inputs into the decision, including, potentially, the partner’s opinion. Prior vaccine acceptance and trust can also be a determinant of vaccine decision-making but was not included as a key component of the 5-P model. Overall, vaccination during pregnancy is a complex decision. Unique pregnancy-specific factors must be considered in addition to other factors applicable to the general population in understanding the decision-making process.

The figure illustrates the key components of the 5-P model (colored wedges) and how the information (pros/cons) arising from these components must be weighed for the mother and fetus. Other factors informing the decision to vaccinate in pregnancy are also shown that are not included in the 5-P model but may be important determinants in some populations.

The factors defining the 5-P model exist within a background of social determinants of health, including proficiency in the native language, geographic location (country, state, neighborhood), education, healthcare access, structural barriers to vaccination, religion, and culture. Historical events also play a role in determining an individual’s underlying trust in medicine, including medical racism and the exclusion of pregnant individuals from vaccine trials. Finally, the pregnant individual must weigh the pros and cons of all these factors for herself and the fetus, resulting in a complex decision-making process. Maternal vaccine decision-making is influenced by these determinants of health, and the factors themselves may change based on the setting. For example, a woman in a low-resource setting may face barriers to accessing adequate healthcare or vaccines and may experience a provider–patient relationship with a community health worker rather than a doctor or nurse.

### 4.1. Perceived Information Sufficiency

Pregnancy is an intense information-seeking phase [52]. Information trusted by pregnant women influences health decisions, behavior, and vaccine acceptance [53]. Information sufficiency is also shown to positively influence pregnancy satisfaction [54]. It has also been shown that for many women, vaccine decision-making for the infant commences during their pregnancy [52]. Given the heightened risk to pregnant women that COVID-19 presents, numerous studies have shown that pregnant women are highly engaged in information-seeking behavior related to COVID-19 infection and vaccines [53]. Conversely, in a study of COVID-19 vaccine uptake during pregnancy, the most cited reason for hesitancy was a lack of knowledge about the vaccine [31]. Lack of data on the safety and efficacy of the vaccine for pregnant women, combined with inconsistent messaging in government advice and guidelines, contributed to vaccine hesitancy and lower vaccine uptake among pregnant women [55]. Exclusion of pregnant individuals from vaccine trials and misinformation about the vaccine’s effects on the fetus and fertility are further barriers to obtaining information regarding pregnancy (Table 2). Pregnant and breastfeeding individuals have lacked the same assurances from clinical research as the general population, leaving them to make vaccination choices without reliable data on the associated risks to themselves and their fetus or child due to the absence of evidence-based data [56].

### 4.2. Pregnancy Protection of the Mother and Fetus (Harm Avoidance)

During pregnancy, there is a philosophical shift in which the pregnant individual prioritizes avoidance of harm to the fetus. This can be beneficial, such as using pregnancy and the responsibility for reducing harm to the fetus as the impetus to stop smoking and other substance use (Table 2) [42]. It can also be potentially detrimental, such as in the case of self-discontinuation of important medications without the advice or supervision of a physician. The desire to avoid potential teratogenic effects has been the motivating factor for people when discontinuing medications like anti-epilepsy medication and psychotropic medications (Table 2) [43,57].

Harm avoidance for the fetus is also directly related to vaccine hesitancy. In a study of pregnancy in France, researchers found that the main reason for not vaccinating was a greater fear of potential deleterious effects of the vaccine on the fetus than of the COVID-19 disease itself [58]. In a Canadian study, perceived personal risk of COVID-19 was not a predictor of a pregnant person’s vaccine acceptance, indicating that for those who were vaccine-hesitant, perceived risk of the vaccine outweighed perceived risk of the illness [59]. Here, the avoidance of risk trumped any possible vaccine benefit. Understanding the hierarchy of priorities is helpful for understanding vaccine uptake during pregnancy. If there is a question about the potential for the vaccine to harm the fetus, the pregnant individual may simply avoid vaccination.

### 4.3. Provider–Patient Relationship

Recommendations from healthcare providers are commonly cited as a reason for vaccine uptake among pregnant people. The term healthcare provider encompasses a vast definition, which may include physicians, nurses, midwives, or community health workers, depending on the setting and resources available. Studies of influenza and Tdap vaccination show that vaccine uptake among pregnant women is higher for those who receive vaccine recommendations from their care providers [60,61]. Pregnant people indicate that they are more willing to receive vaccines if they received a recommendation from their provider to do so [62]. In one study, participants who were ranked as highly likely to receive the COVID-19 vaccine identified physicians to be their main source of COVID-19 vaccine information, and 37% indicated that a physician recommendation would reduce their vaccine hesitancy [63].

In a study on how mothers’ perceptions of vaccines evolved over time, those who experienced increased confidence in vaccines did so in part due to positive relationships and experiences with care providers. Participants described having conversations with trusted doctors, physicians, and public health nurses about the safety and importance of vaccines. Participants reported that convincing techniques used by these care providers included listening to the mothers’ specific concerns, communicating shared values, sharing their (the provider’s) own personal experiences with vaccines, and addressing specific concerns about the ingredients of vaccines or their side effects [64]. We propose that the relationship between a pregnant woman and her obstetric or primary care provider is a critical piece in the decision to vaccinate during pregnancy and enables the pregnant woman to feel empowered to make a decision.

In contrast to the data on the importance of care providers in the pregnant woman’s decision to vaccinate, there are mixed results on the influence of partners and family. A systematic review of studies conducted in low- and middle-income countries representing a vast geographical region found that family members and husbands influenced the decision-making process, with husbands being the most influential individuals [65]. However, a study conducted with pregnant women from Latin America found many women received vaccinations without consulting family or friends, and only a few considered the opinions of family members or male partners [66]. Although the influence of partners and family is important, the opinion of healthcare providers, whether that be physicians or community health workers, appears to supersede that of family and partner as a key determinant for vaccine uptake.

### 4.4. Perceived Benefits

Studies and research have shown that the COVID-19 vaccine provides numerous benefits to pregnant women, including a reduced risk of hospitalization and death for the mother, and transmission of neutralizing antibodies to the fetus that confers disease protection after birth [60,67]. Pregnant individuals with beliefs that the COVID-19 vaccine carried benefits were less vaccine-hesitant and more likely to be vaccinated against COVID-19 or report an intent to vaccinate [36,68,69]. In a Canadian study, self-protection was identified by 65% of respondents as their primary reason for receiving the COVID-19 vaccine [59]. A cross-sectional survey among pregnant women in three U.S. cities found that protection of the fetus/newborn was the most common reason for vaccine acceptance among those who reported intent to receive the COVID-19 vaccine [70]. A study on messaging for the seasonal influenza vaccine directed toward pregnant women found that an emphasis on vaccine benefits for the fetus had a positive impact on maternal vaccine confidence [71]. Researchers have called for research on vaccines for pregnant and breastfeeding women to be prioritized, as data on the safety and efficacy of the vaccine during both pregnancy and breastfeeding would likely contribute to the perceived benefits of the vaccine and, ultimately, vaccine uptake [72].

### 4.5. Perceived Susceptibility and Severity of Disease in Pregnancy

The perception of contracting COVID-19 has fluctuated over time and was greater early in the pandemic. The perception of contracting the disease plays an important role in the decision to get vaccinated and is further complicated by whether they believe a vaccine will provide effective protection [34]. The perceived severity of COVID-19 disease for a pregnant individual is two-pronged, as it involves how both the mother and the infant will be affected. In both cases, a combined perception of the consequences of disease on both individuals plays into the decision on whether contracting the disease will be more or less harmful than getting the vaccine. For COVID-19, the perceived threat of infection was not a direct indicator of willingness to receive the vaccine, as those who believed COVID-19 to be a serious threat were not always willing to receive the vaccine [59]. Multiple studies have shown statistically significant differences in perceived severity of disease between those who get vaccinated and those who do not [30,73]. Interestingly, in one study, perceived threat was not a significant indicator of COVID-19 vaccination hesitancy, suggesting that a potential lack of education, awareness, or perhaps even lack of belief in the consequences of COVID-19 disease may contribute to vaccine resistance [36]. One study found that unvaccinated participants were less likely to agree with the statement that COVID-19 infection is more severe during pregnancy versus their vaccinated counterparts (47% vs. 78%) [30]. These studies suggest that some individuals may be complacent or misinformed about how COVID-19 infection presents differently and more dangerously during pregnancy [30,36]. There are many factors at play that may be contributing to the lack of understanding or belief around the severity of COVID-19 in pregnancy. Two core contributors appear to be a lack of evidence-based communication and disparity in education. A large amount of anecdotal information can be found on social media and TV ads, and individuals who rely on social media platforms such as Twitter or Facebook for health-related information are less likely to receive the influenza vaccine. Furthermore, social media platforms appear to disseminate information regarding vaccine hesitance more successfully than vaccine uptake [74], whereas exposure to direct conversations between healthcare providers and patients has been shown to reduce vaccine hesitancy overall [74]. A wider lens of educational disparities across regions can be used to understand the lack of understanding around the severity of COVID-19 disease for pregnant people. Rates of vaccination in 2021 were found to be lower in rural counties than in urban counties, in part explained by lower education rates [75]. Rural residents have less access to specialty doctors or scientific literature, and their health literacy is generally more limited in these communities [76].

### 4.6. Balancing Risks and Benefits for Oneself and Fetus

The ultimate decision to vaccinate is the assessment by the pregnant individual that there is a greater benefit of vaccination for herself and/or her fetus than harm associated with vaccination. Key factors informing this decision include perceived information sufficiency, harm-avoidance philosophy in pregnancy, relationship with a healthcare provider, and perceived susceptibility and severity of disease. Pregnant individuals who were concerned about the perceived potential risks of the COVID-19 vaccine reported that they were less likely to accept the COVID-19 vaccine [36,68,69]. However, in one study conducted outside of the United States, perceived benefits about the COVID-19 vaccine contributed positively to vaccine acceptance and vaccination rates [77]. “Positive perceptions” included evaluations of trust and mistrust in the benefits of the vaccine, beliefs regarding the vaccine causing infertility, and future risks of the vaccine, among others. Studies have, in turn, hypothesized that the type of information that may contribute to an individual’s understanding of the benefits of the COVID-19 vaccine and/or the potential risks of the COVID-19 disease, such as level of education and access to healthcare, may influence a pregnant individual’s acceptance of the COVID-19 vaccine [77,78]. For many women, this means weighing the perceived risks associated with the vaccine with the risks associated with acquiring COVID-19 [52].

The determinants of vaccine decision-making highlighted in the 5-P model also intersect with one another to influence vaccine decision-making. For example, a trusting relationship with a healthcare provider can facilitate conversations that focus on the benefits of vaccination as well as call attention to the susceptibility and severity of diseases such as COVID-19. A patient–provider relationship built on trust will also influence a pregnant individual’s perception of information sufficiency to encourage them to be vaccinated. This, in turn, impacts an individual’s perception of harm avoidance, as pregnant individuals will be able to determine whether vaccination or taking the risk of not vaccinating (and acquiring the disease) is best for pregnancy protection.

### 4.7. Limitations of Models of Vaccine Acceptance in Pregnancy and Research Gaps

While the HBM includes modifying factors as a core part of the likelihood of action, the assessment of intersectionality among pregnant and recently pregnant individuals is crucial to develop strategies that address COVID-19 vaccine hesitancy. Previous research has demonstrated the need for gender-based approaches to improving vaccine confidence and uptake that also intersect with socioeconomic status [79]. Specifically, more attention and resources toward increasing vaccine uptake should be directed to women living in low-income households and/or working women. Further, culturally competent approaches should be prioritized among African American populations who face higher rates of maternal mortality. Research on vaccine hesitancy should be conducted broadly among minority and non-English speaking groups, who have been historically harmed or marginalized by medicine and science. Research should also be conducted to learn about the reaction of pregnant individuals to social media ads and other public health materials to determine the best method of addressing pregnant women’s unique sources of vaccine hesitancy.

## 5. Conclusions

Increasing vaccine uptake in populations vulnerable to disease is an important goal of public health, but a framework for vaccine decision-making in pregnancy has been lacking and is complex. Pregnant women face unique health concerns, and addressing vaccine hesitancy requires a different approach in public health campaigns to increase vaccine uptake in this unique population. Our 5-P model draws on the HBM [23,24,25], features of decision-making in pregnancy [35], clinical practice, and our research from the One Vax Two Lives social media campaign [21,22] to incorporate factors unique to pregnancy that may be helpful in designing research studies or public health communication campaigns. The 5-P model is an important beginning for further research into the factors that lead to vaccine hesitancy or acceptance in pregnancy.

## Figures and Tables

**Figure 1 vaccines-11-01248-f001:**
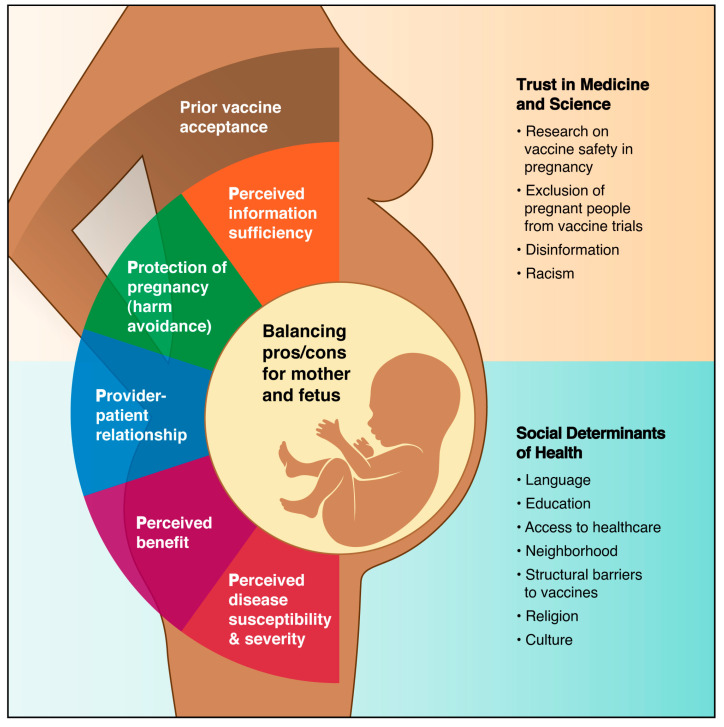
The 5-P model for vaccine decision-making in pregnancy.

**Table 1 vaccines-11-01248-t001:** Factors, strengths, and limitations of vaccine-hesitancy models.

	Health Belief Model [23,25]	3Cs [26]	4C [27]	5A [28]
Is it feasible to obtain the vaccine?	**	Convenience	Convenience	Access and Affordability
Is the vaccine safe?	Barriers	Confidence	Confidence	Acceptance
Is vaccination a priority?	Self-efficacy	Complacency	Complacency	Activation
Is the vaccination important?	Susceptibility, severity, and benefits	**	Utility Calculation	Awareness
Model strengths	Broadly applicable to different populations and health behaviors	Simplicity of model	Simple while addressing worth of vaccination	Addresses worth and safety of vaccine
Model weaknesses	Not specific to vaccine hesitancy or vaccine hesitancy during pregnancy	Lacks consideration of whether the vaccine is deemed valuable	Under-emphasis of the perceived safety of the vaccine	Over-emphasis of access and affordability, less applicable in pregnancy

** The model does not incorporate this element.

**Table 2 vaccines-11-01248-t002:** Examples of factors impacting vaccine acceptance in pregnancy in the 5-P Model.

Factors	Examples
Pregnancy-Specific Factors	
Perceived Information Sufficiency	- Vaccine trials have historically excluded pregnant individuals, making it more difficult for people to feel that they can make informed decisions about vaccination during pregnancy. - A large amount of misinformation exists about the COVID-19 vaccine regarding pregnancy including rumors of miscarriage, birth defects, and future infertility risks [41].
Pregnancy Protection (Harm Avoidance)	- Pregnancy is seen as a window of opportunity to reduce substance use because motivation is high to avoid harm to the fetus [42]. - Pregnancy is also a time that patients self-discontinue medications that they believe may be harmful to the fetus, often without informing their physicians. One study showed high rates of discontinuing anti-epilepsy medications out of fear of teratogenicity [43].
Provider- Patient Relationship	- Providers offer guidance to patients on decisions that impact health like taking prenatal vitamins, tobacco cessation, and vaccinations. - ACOG currently recommends monthly prenatal visits until week 28, biweekly visits until week 36, and then weekly visits until birth [44]. This represents a plethora of opportunities to build relationships and discuss healthy behaviors. This is in contrast to a non-pregnant individual, who average less than 4 medical appointments annually [45]. - Studies in pregnancy find that the obstetric care provider or community health worker recommendations are key factors in decision-making [46,47,48].
Perceived Benefits During Pregnancy	- Maternal–fetal transmission of antibodies during fetal life and after birth via breastmilk.
Perceived Disease Susceptibility and Severity in Pregnancy	- The pregnant individual considers the perceived susceptibility of the maternal–fetal unit to COVID-19, the severity of the disease, and the protection that the vaccine offers. - Some believe that they are more susceptible due to pregnancy and that vaccinating against COVID-19 will provide some protection to the vulnerable fetus. - Others believe that they are young and healthy, and do not perceive a great risk from COVID-19.
**General Factors**	
Prior Vaccine Acceptance	- In the general population, prior acceptance of the influenza vaccine positively predicted acceptance of the COVID-19 vaccine [49].
Social Determinants of Health (SDOH)	- SDOH, including religion, culture, racism, crime and violence, environmental conditions, poverty, education levels, language spoken, and housing stability have been shown to contribute to health disparities in pregnancy [50]. - Black women are more likely to experience pregnancy-related mortality, preterm labor, and infant mortality than non-Hispanic white women. Black women are also more likely to report having experienced poor treatment during obstetric care [51].
Trust in Medicine and Science	- Medical racism has led to less trust in the healthcare system - Historical instances of medical racism, like the Tuskegee Syphilis Study, may underpin vaccine hesitancy.

## Data Availability

No data are associated with this manuscript.

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
