# Peer review of "A Mother’s Dilemma: The 5-P Model for Vaccine Decision-Making in Pregnancy"

_vaccines, 2023, doi:10.3390/vaccines11071248_

Round 1

Reviewer 1 Report

GENERAL

-          The issue is important, the view of the authors is very good, and the model is excellent

-          In the model itself, the author mentioned the components of the model, but I suggest the additional explanation regarding the correlation among those components. Usually, the model will also mention these relationships. Which factor influence other factors? Was it two ways or only one way?

-          Are there any differences if the model will be implemented in low resource setting?

-          What about the influence of the culture, and religion (especially in certain parts of the world)?

-          And what about the influence of family or extended family, which become a common issue in low resource settings?

TITLE

-          Didn’t the author have a name for the model?

-          Why the author did not mention about the model in the title?

REFERENCES

-          Page 1: Line 38-39; I suggest number “1,2,3,4,5,6,7,8,9,10,11,12,13,14,15” to be written as “1-15”

-          Some references (such as number 9,11,14,15) were not written properly

Author Response

GENERAL

  • The issue is important, the view of the authors is very good, and the model is excellent.

Response: We appreciate the Reviewer’s comments.  We agree and feel that this model is needed to make progress in targeting health communication campaigns to pregnant people.

  • In the model itself, the author mentioned the components of the model, but I suggest the additional explanation regarding the correlation among those components. Usually, the model will also mention these relationships. Which factors influence other factors? Was it two ways or only one way

Response: We have now discussed relationships among factors, specifically how healthcare providers influence the other components and how information sufficiency can influence harm avoidance.

-- Are there any differences if the model will be implemented in low resource setting?

Response: Thank you for your question. Yes, there are differences in that low resource settings will be most impacted by social determinants of health, such as access to healthcare and structural barriers to obtaining vaccines. Additionally, a trusting relationship with a care provider may rely more on a relationship with a community health worker rather than a doctor or nurse in a low resource setting. However, we do not believe the main components of the model change; rather, the model is influenced by those social determinants of health. 

The bottom of Section 4, before 4.1 begins, now reads: “Maternal vaccine decision-making is influenced by these determinants of health, and the factors themselves may change based on setting. For example, a woman in a low resource setting may face barriers to accessing adequate healthcare or vaccines and may experience a provider-patient relationship with a community health worker rather than a doctor or nurse.”

  • What about the influence of culture, and religion (especially in certain parts of the world)?

Response: Social determinants of health include non-medical factors that influence health outcomes.  Religion and culture are social determinants of health. We have now included religion and culture in our model figure (under social determinants of health) to highlight these factors.  

  • And what about the influence of family or extended family, which has become a common issue in low resource settings?

Response: Thank you for the suggestion. In reviewing the literature, there evidence to support that health care providers play a greater role in influencing pregnant individuals’ decision to be vaccinated than family members. In several studies in low-resource settings in Africa and Latin America, family has been shown to be less influential in vaccine decision-making than the health care provider or community health workers (PMID: 29859803, PMID: 35081138, PMID: 31319932, PMID: 36173503).

  • Didn’t the author have a name for the model?

Response:  We have now named the model, “The 5-P Model” to encapsulate the main model features: 1) Pregnancy Protection for Mother and Fetus, 2) Provider-Patient Relationship, 3) Perceived Benefit, 4) Perceived Disease Severity/Susceptibility, and, 5) Perceived Information Sufficiency.  

  • Why the author did not mention about the model in the title?

Response: This is an excellent suggestion.  We have now named the model in the manuscript title, which is revised to, “A Mother’s Dilemma: the 5-P Model for Vaccine-Decision Making in Pregnancy.”

REFERENCES

-          Page 1: Line 38-39; I suggest number “1,2,3,4,5,6,7,8,9,10,11,12,13,14,15” to be written as “1-15”

-          Some references (such as number 9,11,14,15) were not written properly

Response: Thank you.  We have now fixed the references.

Reviewer 2 Report

The research deals with a topical issue and is well structured, proposing a path for information to women that can be shared. I note the opportunity to integrate the issue of regulatory references; the following bibliographic reference is useful for this purpose 

"The state of Play on COVID-19 Vaccination in Pregnant and Breastfeeding Women: Recommendation, Legal Protection, Ethical Issues and controversies in Italy" https://doi.org/10.3390/healthcare11030328

Author Response

REVIEWER #2:  Comments and Suggestions for Authors

  • The research deals with a topical issue and is well structured, proposing a path for information to women that can be shared. I note the opportunity to integrate the issue of regulatory references; the following bibliographic reference is useful for this purpose: "The state of Play on COVID-19 Vaccination in Pregnant and Breastfeeding Women: Recommendation, Legal Protection, Ethical Issues and controversies in Italy" https://doi.org/10.3390/healthcare11030328

Response: Thank you kindly.  We have now incorporated this reference and a few lines to discuss the content of this article.

Reviewer 3 Report

I read with interest the article “A Mother’s Dilemma: Vaccine Decision-Making in Pregnancy”.

Certainly, SARS-COV-2 raised the issue of vaccination during pregnancy much more than any other infection did before.

The article well describes the issue of vaccine hesitancy and models for vaccine decision-making. We are now approaching a phase of end of COVID-19 pandemic and should take advantage from this experience to build better pregnant women’s confidence in science and vaccine.

Authors could just comment on the existence of many clinical practice guidelines 

(Systematic review and critical evaluation of quality of clinical practice guidelines on the management of SARS-CoV-2 infection in pregnancy. Am J Obstet Gynecol MFM. 2022 Sep;4(5):100654. doi: 10.1016/j.ajogmf.2022.100654. Epub 2022 May 2. Erratum in: Am J Obstet Gynecol MFM. 2022 Aug 5;:100683. PMID: 35504493; PMCID: PMC9057927.) to shortly discuss the “distance” between national and international bodies releasing these documents and the population of pregnant women. Indeed, in the era of social media, knowledge is not anymore considered the prerogative of the few and therefore questionable opinions could be launched and gain diffusion and credit even if not supported by data or critical and scientific rationale.

Please consider that the references should be placed before the full stop at the end of the sentence, e.g. “….. (reference).” and not after it.

In addition, whenever consecutive references are cited in a place, they could be cumulated from first to last, e.g. “1, 3-6, 8” (which means 1-3-4-5-6-8).

Author Response

  1. Certainly, SARS-COV-2 raised the issue of vaccination during pregnancy much more than any other infection did before. The article well describes the issue of vaccine hesitancy and models for vaccine decision-making. We are now approaching a phase of end of COVID-19 pandemic and should take advantage from this experience to build better pregnant women’s confidence in science and vaccine.

RESPONSE: Thank you for this comment. We agree and hope that this model will be helpful for the field moving forward. 

  1. Authors could just comment on the existence of many clinical practice guidelines. (Systematic review and critical evaluation of quality of clinical practice guidelines on the management of SARS-CoV-2 infection in pregnancy. Am J Obstet Gynecol MFM. 2022 Sep;4(5):100654. doi: 10.1016/j.ajogmf.2022.100654. Epub 2022 May 2. Erratum in: Am J Obstet Gynecol MFM. 2022 Aug 5;:100683. PMID: 35504493; PMCID: PMC9057927.) to shortly discuss the “distance” between national and international bodies releasing these documents and the population of pregnant women. Indeed, in the era of social media, knowledge is not anymore considered the prerogative of the few and therefore questionable opinions could be launched and gain diffusion and credit even if not supported by data or critical and scientific rationale.

RESPONSE: This is an excellent suggestion and we have now added these references and language to reflect the communication gap between the pregnant lay public and vaccine experts in pregnancy and their clinical practice guidelines. We feel that medical societies could play a greater role in public health communication campaigns by taking advantage of new research in this space and this model for vaccine decision-making.  

  1. Please consider that the references should be placed before the full stop at the end of the sentence, e.g. “….. (reference).” and not after it.

RESPONSE: Thank you. We have now placed the references before the full stop at the end of the sentence. 

  1. In addition, whenever consecutive references are cited in a place, they could be cumulated from first to last, e.g. “1, 3-6, 8” (which means 1-3-4-5-6-8).

RESPONSE: The revised manuscript is now using the MDPI Endnote Style for Vaccines and has collated the references in line with this comment.  

Reviewer 4 Report

My specific comments on this manuscript are-

1. Fifteen references for first three lines in introduction is too much. It needs to be checked.

2. Vaccination can play a critical role in protecting……It should be crucial not critical.

3. Introduction part is unnecessarily long. It should be concise and brief. Also, the citations are in wrong style.

4. Figure in page 5 must be revised. The human figure should be omitted.

5. Detail methodology should be described.

Author Response

REVIEWER#3:  Comments and Suggestions for Authors

My specific comments on this manuscript are-

  1. Fifteen references for first three lines in introduction is too much. It needs to be checked.

Response: Thank you.  We have now checked and reduced the number of references as requested, but wanted to keep references that spanned a variety of infectious diseases that impact pregnancy (i.e., congenital rubella syndrome, congenital CMV, influenza, COVID-19).

  1. Vaccination can play a critical role in protecting……It should be crucial not critical.

Response: Thank you, we have now changed this word to crucial.

  1. Introduction part is unnecessarily long. It should be concise and brief. Also, the citations are in wrong style.

Response: We have now made the introduction briefer. We have now formatted the citations in the MDPI style, which we believe is appropriate for the Vaccines journal. 

  1. Figure in page 5 must be revised. The human figure should be omitted.

Response: We respectfully request to keep the human figure, as we feel that this helps to show the reader which components are clearly maternal, fetal and societal.  We feel that it is these separate components of vaccine decision making that differentiate the pregnant versus non-pregnant decision to vaccinate.

  1. Detail methodology should be described.

Response: We have now added additional citations to describe our methodology, which was informed by literature review, prior models and our own experience.

Round 2

Reviewer 4 Report

no comments

no comments

Author Response

We thank Reviewer #4 for your review and assessment of our manuscript. We note that you would like minor editing of the English language, which we have now provided in the revised manuscript.  Thank you again for your review.